# Expanding the Phenotype of the CACNA1C-Associated Neurological Disorders in Children: Systematic Literature Review and Description of a Novel Mutation

**DOI:** 10.3390/children11050541

**Published:** 2024-04-30

**Authors:** Lorenzo Cipriano, Raffaele Piscopo, Chiara Aiello, Antonio Novelli, Achille Iolascon, Carmelo Piscopo

**Affiliations:** 1Department of Molecular Medicine and Medical Biotechnology, University Federico II, 80131 Naples, Italy; lorenzo.cipriano@unina.it (L.C.); achille.iolascon@unina.it (A.I.); 2Department of Neuroscience, Reproductive and Odontostomatological Sciences, University Federico II, 80131 Naples, Italy; raffaele.piscopo2@unina.it; 3Laboratory of Medical Genetics, Translational Cytogenomics Research Unit, Bambino Gesù Children Hospital, IRCCS, 00146 Rome, Italy; chiara.aiello@opbg.net (C.A.); antonio.novelli@opbg.net (A.N.); 4Medical and Laboratory Genetics Unit, A.O.R.N. “Antonio Cardarelli”, 80131 Naples, Italy

**Keywords:** Timothy syndrome, CACNA1C gene, intellectual disability

## Abstract

**Background:** CACNA1C gene encodes the alpha 1 subunit of the CaV1.2 L-type Ca2+ channel. Pathogenic variants in this gene have been associated with cardiac rhythm disorders such as long QT syndrome, Brugada syndrome and Timothy syndrome. Recent evidence has suggested the possible association between CACNA1C mutations and neurologically-isolated (in absence of cardiac involvement) phenotypes in children, giving birth to a wider spectrum of CACNA1C-related clinical presentations. However, to date, little is known about the variety of both neurological and non-neurological signs/symptoms in the neurologically-predominant phenotypes. **Methods and Results:** We conducted a systematic review of neurologically-predominant presentations without cardiac conduction defects, associated with CACNA1C mutations. We also reported a novel de novo missense pathogenic variant in the CACNA1C gene of a children patient presenting with constructional, dressing and oro-buccal apraxia associated with behavioral abnormalities, mild intellectual disability, dental anomalies, gingival hyperplasia and mild musculoskeletal defects, without cardiac conduction defects. **Conclusions:** The present study highlights the importance of considering the investigation of the CACNA1C gene in children’s neurological isolated syndromes, and expands the phenotype of the CACNA1C related conditions. In addition, the present study highlights that, even in absence of cardiac conduction defects, nuanced clinical manifestations of the Timothy syndrome (e.g., dental and gingival defects) could be found. These findings suggest the high variable expressivity of the CACNA1C gene and remark that the absence of cardiac involvement should not mislead the diagnosis of a CACNA1C related disorder.

## 1. Introduction

L-type calcium channels are widely expressed in the human body and contribute to the physiological functioning of numerous systems such as heart, brain, smooth muscle, and the immune system [1]. The main function of these channels is to control the calcium transit through the cell membrane, performing the function of modulating the action potential and handling several secondary signaling pathways. To date, four different subtypes of L-type channels have been described: CaV1.1, CaV1.2, CaV1.3 and CaV1.4. Of them, CaV1.2 shows the largest expression across multiple tissues including the heart, brain, smooth muscle, endocrine and immune system [2]. Therefore, the disruption of this channel (and, in general, of the L-type channels) have a relevant consequence on multiple systems and tissues. In the heart, CaV1.2 channels activate the ryanodine receptor, promoting the calcium release from the sarcoplasmic reticulum [3] and shaping the cardiac action potential [4]. In the brain, CaV1.2 channels are highly expressed, representing around 90% of all the L-type calcium channels [5], and are located in multiple cerebral regions (especially in the hippocampus) and in the cerebellar cortex. The high expression in some of these regions explain the important role of CaV1.2 channels in neuronal plasticity and long-term potentiation (and, therefore, in learning and memory) [6,7]. CaV1.2 channels are also highly expressed in chondrocytes and osteoblasts, being essential for normal mandibular development [8,9], probably explaining some peculiar phenotypical features in patients affected by CaV1.2 channel dysfunction (e.g., dental malformations and other musculoskeletal anomalies).

L-type calcium channels are encoded by CACNA1S, -C, -D and -F genes; the CACNA1C gene encodes the alpha 1 subunit of the CaV1.2 L-type Ca2+ channel. Pathogenic variants in the CACNA1C gene have been historically related to cardiac rhythm disorders. In detail, the three principal cardiac (or mainly cardiac) phenotypes are long QT syndrome, Brugada syndrome and Timothy syndrome, a complex association of heart structural and conduction defects with dysmorphic facial features, syndactyly, intellectual disability, epilepsy and autism [8]. However, in the last few years a growing number of studies have described the association between CACNA1C mutations and brain impairment, even in absence of cardiac involvement, both in pediatrics and adults patient [10,11]. Indeed, pathogenic variants in this gene have been related to psychiatric disorders such as schizophrenia, bipolar disorder and major depressive disorder [12] as well as to several neurological features including ataxia, epilepsy, intellectual disability, hypotonia [10]. In particular, in genome-wide association studies, CACNA1C emerged as the most robustly replicated psychiatric risk gene in major depression, bipolar disorder, schizophrenia, and autism [13,14,15,16,17,18,19]. Individuals with selected CACNA1c single nucleotide polymorphisms (mainly rs1006737) showed an increase in multiple traits including psychiatric features such as depression, anxiety and startle reactivity, and also neurological as decreased verbal fluency [20,21,22]. In a recent study, Levy et al. found that CACNA1C-related disorders (mainly Timothy syndrome 1 and 2) showed an increased prevalence of neuropsychiatric symptoms as compared to healthy controls [23]. In detail, they showed a high prevalence of neurological and/or psychiatric symptoms such developmental delay (in 92% of cases), incoordination (in 71% of cases), autism spectrum disorder (in 50% of cases) and also seizures (in 37.5%). Considering the neurologically isolated presentation, CACNA1C mutation were recently found in a family with ataxia [24].

As a consequence, there is a now an increased interest for neurologic and psychiatric signs and symptoms in the common phenotype of the CACNA1C related disorders.. However, a lot it remains to know about the behavioural and neurological impairment in non cardiac CACNA1C related disorders in children, such as the spectrum of neurological manifestations associated with CACNA1C mutations, and still it is unknown if a dysmorphic features typical of the Timothy syndrome can be associated.

In the view of answering these questions, this study aims to perform a systematic review of published data concerning CACNA1C mutations associated with psychiatric/neurological isolated presentations in children and adults patients. We also describe a novel de novo missense CACNA1C variant associated with predominant psychiatric and neurological features, in absence of cardiac conduction defects.

## 2. Methods

### 2.1. Systematic Review

We conducted a systematic review of neurologically-predominant presentations without cardiac conduction defects, associated with CACNA1C mutations. The study was performed according to the Preferred Reporting Items for Systematic Reviews and Meta-Analyses (PRISMA) guidelines [25].

The eligible studies were identified through a comprehensive literature search in PubMed from 1 January 1900 to 8 February 2024. Cohort, case-control and cross-sectional studies as well as case series and single case reports were all included. The search strategy encompassed the use of Boolean operators on the following terms “CACNA1C”, “neurology”, “neurological”, “psychiatric”, “neuropsychiatric”, “ataxia”, “intellectual disability”, “epilepsy” “development”, “schizophrenia”, “bipolar disorder”, “depression”, “anxiety”, “autism”, “ADHD”, “hypotonia” and “apraxia”. We also screened the references listed in all the relevant studies including research articles, meta-analyses, and reviews. Two reviewers (LC and CP) assessed the eligibility of the identified studies. The process was divided in two phases. The first part consisted of a screening of titles and abstracts of the selected articles. The second step was characterized by a full-text review. The reviewers (LC and CP) performed both the steps independently. In case of disagreements on study selection the choice of retaining or not including the article was based on the opinion of a third reviewer (RP). The following inclusion criteria were considered for the present study: (1) studies performed on human subjects only; (2) cohort, case-control, and cross-sectional studies, case reports, case series and literature review including CACNA1C mutations; (3) studies that provided detailed genotype information; (4) studies that provided extensive information about phenotype; (5) studies published in english language only. Exclusion criteria were: (1) studies that provided insufficient information about the mutation type and/or a minimal amount of information about the demographic and clinical presentation characteristics; (2) studies including Copy number variation (CNV) involving more genes than CACNA1C; (3) studies including CACNA1C pathogenic variants associated with documented cardiac conduction defects. The extracted data included the details about the CACNA1C variant, including type (missense/truncating) and inheritance (de novo/inherited), neurologic and psychiatric characteristics and all the other relevant clinical features including all the typical signs or symptoms commonly associated with the Timothy syndrome.

The risk of bias was assessed through the Critical Appraisal Checklist for both case reports and case series, developed and validated by the Joanna Briggs Institute (JBI, Appendix A). The JBI checklist for case reports include a total of eight questions evaluating the precision and the reliability of the case report description and presentation on the basis of the information provided by the authors (demographic characteristics, familial and personal history, respect for diagnostic criteria, treatment etc.). The JBI checklist for case series is comparable with the checklist for case reports with a greater interest in inclusion criteria and statistical analyses, for a total of ten questions. The researchers can answer ‘yes’, ‘no’, ‘unclear’ or ‘not applicable’ in response to each item. The greater the number of ‘no’ or ‘unclear’ selected, the greater the risk of bias in each category and in each study. This step was also carried out by the two reviewers (LC and CP) independently, always in reliance on the opinion of a third researcher (RP) in case of disagreements.

A spreadsheet was adopted as a data extraction tool. Statistical analyses were not performed due to the small sample size and the frequency of each clinical feature was calculated through the R software (http://www.rstudio.com, version 4.3.2, accessed on 2 February 2024).

### 2.2. Genetics/Materials and Methods (Clinical Exome)

Clinical investigations and genetic analyses were conducted in accordance with the Helsinki Declaration. After obtaining informed consent from the patient’s parents for the genetic testing, peripheral blood samples were collected from the affected subject and the unaffected parents and genomic DNA was extracted from circulating leukocytes. Enrichment and parallel sequencing were then performed utilizing a custom clinical exome panel, containing more than 8500 genes. Library preparation was carried out by using the Twist enrichment kit, according to the manufacture’s protocol (Twist Bioscience, South San Francisco, CA, USA), and sequenced on a NovaSeq 6000 (Illumina, Inc., San Diego, CA, USA) platform. The BaseSpace pipeline (Illumina, https://basespace.illumina.com/) and the Geneyx software (LifeMap Sciences, https://geneyx.com/) were used for the variant calling and annotating variants, respectively. Sequencing data were aligned to the hg19 human reference genome, with an average alignment coverage over target region of the custom clinical exome panel (Twist enrichment kit) of 289.21X and of 281X for the c.4513G>A variant in the CACNA1C gene. All the identified variants were analyzed in silico by using Combined Annotation Dependent Depletion (CADD) V.1.3, Scale-Invariant Feature Transform (SIFT), Polymorphism Phenotyping v2 (PolyPhen-2) and Mutation Taster for the prediction of deleterious non-synonymous SNVs for human diseases. Variants were examined for coverage and Qscore (minimum threshold of 30), and visualized by the Integrative Genome Viewer (IGV). DRAGEN (Dynamic Read Analysis for GENomics) (Illumina, Inc., San Diego, CA, USA) and the TGex software (LifeMap Sciences, http://tgex.genecards.org/) were used for the variant calling and annotating variants, respectively.

Clinical exome sequencing performed on the patient and his parents enabled to identify a novel heterozygous de novo variant c.4513G>A in the CACNA1C gene (NM_000719.7), causing the amino acidic change p.Asp1505Asn. The variant was confirmed by sanger sequencing. It has never been described before, and can be classified as a likely pathogenic variant (class 4) according to the ACMG criteria PM2 (absent from controls in Exome Sequencing Project, 1000 Genomes Project, or Exome Aggregation Consortium, variant allele frequency: 0), PP3 (multiple lines of computational evidence support a deleterious effect on the gene or gene product, CADD score 27.3), PP2 (missense variant in a gene that has a low rate of benign missense variation and in which missense variants are a common mechanism of disease, Z-score: 7.27, https://gnomad.broadinstitute.org/gene/CACNA1C, accessed on 2 February 2024), PS2 (de novo in a patient with the disease and no family history, de novo) and according to the Combined Annotation Dependent Depletion score (CADD score 27.3). This variant is not present in any database (dbSNP) [http://www.ncbi.nlm.nih.gov/projects/SNP] (accessed on 2 February 2024), 1000 Genomes Project, Geno2MP [https://geno2mp.gs.washington.edu/Geno2MP] (accessed on 2 February 2024), gnomAD [https://gnomad.broadinstitute.org] and can be considered as a private variant.

## 3. Results 

### 3.1. Systematic Review

A total of 660 studies from the PubMed database were found. All the studies were screened on the basis of their titles and abstracts, and 22 were retained for full-text analysis, according to our inclusion and exclusion criteria. The reasons for article exclusion are summarized in the PRISMA flowchart (Figure 1). The main reason was an incomplete clinical description. Two articles were excluded due to the involvement of more genes than the only CACNA1C. Five adjunctive studies were added through a cross-referencing process. Of the 27 selected studies for full-text analysis only six were included in our review [10,11,24,26,27,28].

Clinical and genetics features of a total of 35 patients (from five studies) with CACNA1C associated neurological impairment without documented conduction defects were extracted. The cohort was extended with an adjunctive case (a patient with CACNA1C associated neurological predominant phenotype and clinical features of Timothy syndrome without conduction defects) from our genetic department.

All the selected articles were case series, with extremely variable sample size (the main features of the included articles are summarized in the Table 1). The largest casuistry was the one reported by Rodan et al. that included a total of 25 patients from 22 families with heterozygous variants in CACNA1C [10]. According to the previously specified inclusion and exclusion criteria, 22 individuals of this cohort were included, and their characteristics were extracted. The other studies reported small cohorts (up to 5 individuals) with different types of mutations, ranging from the truncating/non-truncating mutations to CNVs, passing by intronic variants. This contributed to the high heterogeneity of the sample, making the investigation of some aspects (e.g., genotype-phenotype correlations) difficult. All the included individuals had heterozygous variants in the CACNA1C gene. Most of the cases presented a de novo variant whereas in a small fraction of them the pathogenic variant was paternally inherited. 

The most frequent neurologic characteristic was a variable aucase of intellectual disability that was found in 89% of the cases (Figure 2), followed by hypotonia (55.5%), ataxia (52%), seizures (37.9%) and a clinical diagnosis of autism (29.2%).

Missense variants were the most represented (44%). Two cases described a CNV (a duplication and a deletion, respectively, [27,28]), including only exons of the CACNA1C gene. 

### 3.2. Case Description

The proband was a 11-years old aucasian male. He was a full term infant with parameters (weight, length, head circumference and Apgar score) at birth in the normal range. His family history was unremarkable. Since the age of 3 years, he was diagnosed with a speech delay associated with abnormal behaviors such as stereotypes and obsessive-compulsive aspects. Ophthalmologic evaluations, auditory evoked potential, brain MRI, abdomen ultrasound exam and blood routine tests were performed; all these investigations were normal. An extensive cardiological evaluation performed at 10 years old, revealed a bicuspid aortic valve and fossa ovalis aneurysm without any heart conduction defect. In terms of neurodevelopmental features, he showed (mainly in the first years of life) repetitive behavior and stereotypy following sensory stimuli, receiving a diagnosis of mild autism spectrum disorder (ADOS-2 score = 8). On the clinical evaluation he was noted hypertelorism with mild bilateral ptosis, flat nasal bridge with upturned and large tip, gingival hyperplasia with abnormal shaped teeth (conical) and diastema, teletelia, pectus carinatum, thoracic kyphosis, prominent fingertip pads, clinodactyly of the fourth and fifth toe, bilaterally, right first toenail dystrophy, bilateral pes planovalgus and joint laxity (Figure 3). The neurological evaluation showed bilateral clonus at lower limbs, oro-buccal-facial hypotonia, constructional, dressing and oro-buccal apraxia and a mild ataxia. Although he never suffered from clinical seizures, the sleep EEGs showed epileptogenic anomalies in the left temporal lobe.

In terms of social interaction, he showed important anxiety and emotional difficulties. He performed well at the Raven’s Progressive Matrices (97° centile), showing a high grade of non-verbal fluid intelligence. 

In the suspicion of a genetic cause of the presented disorder, we performed an extensive genetic investigation. The first genetic examination included a traditional karyotype and FMR1 repeat expansion analysis that did not evidence pathologies. So, an SNP-array was performed without identifying causative CNVs (arr(1-22)x2,(X,Y)x1). Therefore, the little patient and his parents underwent a clinical exome sequencing. This investigation was able to identify a novel heterozygous de novo variant c.4513G>A in the CACNA1C gene (NM_000719.7), causing the amino acidic change p.Asp1505Asn. This variant, never described before, was classified as a likely pathogenic variant (class 4) according to the ACMG criteria (PM2, PS2, PP2 and PP3) and according to the Combined Annotation Dependent Depletion score (CADD score 27.3). Also, the variant is not present in any database (dbSNP) [http://www.ncbi.nlm.nih.gov/projects/SNP] (accessed on 2 February 2024), 1000 Genomes Project, Geno2MP [https://geno2mp.gs.washington.edu/Geno2MP] (accessed on 2 February 2024)], gnomAD [https://gnomad.broadinstitute.org] and can be considered as a private variant. The clinical exome sequencing also revealed a maternally inherited c.3154C>T (p.Arg1052Ter) variant in the MYH6 gene. This variant was classified, according to the ACMG, as likely pathogenic.

## 4. Discussion

We conducted a systematic review of the literature on *CACNA1C* reported cases with brain impairment in absence of cardiac conduction defects. We also expanded the resultant cohort with a description of a novel de novo missense variant in the *CACNA1C* gene in a patient with predominant psychiatric and neurological manifestations.

Intellectual disability was the clinical feature with the highest prevalence (89.2% of the cases). However, it should be pointed out that the degree of intellectual disability appeared quite variable (in a large range from mild to severe) and only few studies [10,24] reported detailed evaluation of cognitive functions. The high prevalence of intellectual disability in *CACNA1C* related disorders is explained with the role played by the calcium ion channels in both prenatal and postnatal brain development. In fact, the calcium-facilitated depolarization regulates neural proliferation, migration, and differentiation during the formation of the cerebral cortex and, postnatally, it contribute to the modeling of the synapses and sensory neural circuits [29,30]. 

Ataxia, a neurological sign defined as impaired coordination of voluntary muscle movements, is mainly due to abnormalities in cerebellar efferent and afferent pathways. The role of the calcium channels in cerebellar pathways has been widely described [31] as well as it is commonly agreed the association between calcium channelopathies, mainly N and T calcium channels (respectively encoded by the genes CACNA1A and CACNA1G-H-I), and cerebellar ataxia [32]. Moreover, mRNA of L-type CaV1.2 channels are highly expressed in the cerebral cortex, the pituitary gland, the amygdala, the basal ganglia, and also in the cerebellum [8].In mouse models of ataxia and epilepsy, cerebellar calcium channels such as Cav1.2 were significantly reduced in number [33]. To date, only one case describing an isolated ataxic syndrome associated with a CACNA1C mutation, has been reported in humans [24].

Calcium channels are also involved in epilepsy and numerous drugs, commonly used in epilepsy treatment, exert their function in controlling or preventing seizures through their action on calcium channels. Although T-type channels are the most frequently associated with seizures, the relationship between L-type CaV1.2 and febrile seizures has also been reported [34]. 

Concerning the abnormal behaviors as well as the impaired social communication and interaction, Liao et al. recently highlighted the crucial role of the genetic variants encoding voltage-gated calcium channels in the pathogenesis of autism spectrum disorders [35]. 

Our patient presented dressing, constructional and speech apraxia, mild intellectual disability, EEG abnormalities and bilateral ankle clonus. Apraxia is a neurological disorder that affects the ability to perform everyday movements due to defects in motor skill, planning, conceptualization and use of tools and/or knowledge of actions/series of actions, in the absence of relevant damage to motor or sensory pathways. The type of apraxia can usually address the clinician to a specific neuroanatomical correlation. Dressing apraxia is due to damage of the dorsal visual association cortex and pathways in the parietal lobe that impair attentional, spatial and kinesthetic analyses [36]. It is usually associated with right parietal lesions. Constructional apraxia is the inability to copy the spatial pattern in which things are arranged; it has been linked to parietal lesions in the left and right parietal hemisphere [37]. Apraxia of speech affects the motor programming system for speech production with a resultant inability to translate conscious speech plans into motor plans; it has been associated with left frontal lobe dysfunction [36]. The contemporaneous presence of multiple apraxia types makes it impossible to identify a unique neuroanatomical correlation. This complex apraxic phenotype is probably the result of a global dysfunction in brain development and function.

Apraxia of speech has been previously described in only one case of a 5-years old male patient with a 2.3 Mb de novo 12p13.33-p13.32 deletion, the patient also showed intellectual disability and behavioural problems [38]. However, this 12p13.33-p13.32 deletion encompassed more than the only CACNA1C gene. In general, apraxia is not commonly included in the CACNA1C-related neurologic features, even in the most recent descriptions of the CACNA1C-associated disorders. 

The current literature provides little information about the role of the mutation type (missense or truncating) in the genesis of the neurological and psychiatric disturbances. Rodan et al., showed an apparent phenotypic difference between individuals with nontruncating and truncating variants in their cohort [10]. They found that nontruncating variants mainly occurred de novo and showed a more severe neurological impairment, especially in terms of intellectual disability and epilepsy. The truncating variants were more commonly (as compared to the non-truncating ones) familial and showed milder neurological phenotypes, usually characterized by expressive language defects, autism spectrum disorders and an overall less severe developmental delay. The reason for these differences remains unexplained and also patch clamping studies were not able to clarify these aspects [10]. Our variant in the CACNA1C gene is located in the S6 transmembrane domain (of the IV domain) of the protein and its position suggests a possible implication in modulating the calcium current. A previously reported (and also studied with patch clamping methods) CACNA1C variant, near our variant (located in the S5 transmembrane domain of the IV domain) showed a clear effect on channel activity determining a reduction in calcium current [10]. In general, both CACNA1C gain and loss of function can determine severe neurological and psychiatric symptoms [10] whereas Timothy syndrome is commonly a result of a gain of function [39]. So, one might speculate that reduction in calcium conductance, determined by some missense variants, can cause neurological and psychiatric characteristics commonly found in the Timothy syndrome without cardiac involvement whereas a gain of function with increased calcium conductance could determine the association of neurological and cardiac impairment typical of the Timothy syndrome. Concerning the difference in neurological phenotype severity between truncating and non-truncating proteins, a possible explanation could reside in the fine equilibrium between excitation and inhibition that the brain needs for a proper functioning. In fact, both gain and reduction in calcium conductance can easily dysregulate the imbalance of neuronal excitation and inhibition that could favor seizure and hinder long-term potentiation (the basis of learning, memory and in general the high cognitive functions). This could also explain why the haploinsufficiency determined by the truncating mutations seems to be better tolerated whereas missense variants altering channel conductance show more frequently a dominant negative effect.

Concerning the non neurological clinical features, commonly reported in the Timothy syndrome, syndactyly was the most frequently described (9.7% of the cases). Our case showed a gingival hyperplasia and abnormal tooth shape. Although dental defects as well as gingival hyperplasia have been reported in few previous CACNA1C case descriptions, these clinical characteristics have been found only in association with Timothy syndrome phenotype [40,41,42]. Ion channels are well known for being involved in tooth formation [43], additionally, defects in gene encoding L-type calcium channels have been associated with abnormal dentition [44,45]. To the best of our knowledge, in the present study we describe for the first time dental and gingival defects associated with a neurological isolated CACNA1C disorder.

Gingival overgrowth is a frequently reported complication of calcium channel blocker. Our mutation, located in proximity to a previously described variant inducing reduction in calcium conductance [10], may determine a decrease in calcium transit and so, favor the gingival growth. Why these dental and gingival anomalies are reported only in association with the typical Timothy syndrome and have never been described in the neurologically isolated CACNA1C disorders is still unexplained. This could represent a real difference (e.g., based on the hyperactivation/hypoactivation of the channel according to the mutation type, i.e., gain or loss of function) or a detection bias (i.e., less attention was paid on this defect). Future studies on larger samples of neurologically isolated CACNA1C-related presentation are needed to solve this dilemma.

The wide variability of these clinical presentations could depend on the CACNA1C gene and its property. In fact, CACNA1C gene undergoes alternative splicing and some of its transcripts are subjected to additional RNA editing, so determining more than 200 unique CACNA1C transcripts in the human brain, that can possibly encode L-type calcium channels with different function and kinetics [46]. The differences in the abundance of CACNA1C transcripts between brain regions could explain the heterogeneity in CACNA1C-related neurological presentations. It is noteworthy that the splicing profile varies between the different brain areas and, in particular, across the different regions of the cerebellum; this could influence the prevalence and the severity of the ataxic symptoms in the affected patients [46]. Also the severity of intellectual disability and the presence or not of seizures could be affected by this aspect. In fact, not all the brain regions (and the resultant networks) are equally important for superior cognitive functions as well as a small fraction of the brain cortex is composed of eloquent areas (where epileptic focus translates in clinically evident seizures).

The patient also carried a likely pathogenic variant in the MYH6 gene, inherited from his mother. Likely pathogenic and pathogenic variants in this gene have been associated with structural cardiac defects including dilated or hypertrophic cardiomyopathy and atrial septal defect. Our patient presented a bicuspid aortic valve and a fossa ovalis aneurysm; both these conditions are not compatible with the MYH6-associated phenotypes. An extensive cardiac evaluation, comprehensive of cardiac ultrasound, has been suggested to the mother to better interpret the pathogenic role of the variant.

The present work is not exempt from limitations. Firstly, few cases of neurologically isolated CACNA1C gene mutations have been described in literature and only some of them were considered eligible for the systematic review. So, the sample size represents the major limit of the current study. The small sample also made senseless approaches with statistical analyses. Secondly, shortcomings are the possible biases related to the systematic review process; one of them could be considered the bias in detecting specific neurological and non-neurological signs and symptoms. Although clinical features reported in each article were found and described by physicians, it cannot be excluded a different expertise influencing the sensitivity in recognizing some neurological characteristics. Part of the included studies were small case series contributing to a lower homogeneity in the clinical evaluation of the patients. In addition, we did not provide a functional validation of the identified variant. Additionally, without patch clamping studies, we were not able to define the exact role of the variant in the calcium channel conductance.

In summary, the current work expands the phenotype of the CACNA1C neurologically isolated conditions. We included apraxia in the landscape of the neurological signs and symptoms associated with CACNA1C mutations. We also reported the high frequency of ataxia and epilepsy in both children and adult patients. This is an interesting aspect because it motivated the search for CACNA1C mutations in case of clinical presentation of apraxia, ataxia and/or epilepsy, isolated or associated with each other. Identifying a CACNA1C gene mutation could open to new possibilities in terms of precision medicine. In this perspective, L-type calcium channels could be adequate drug targets in CACNA1C-related neurological and psychiatric disorders [47,48].

## 5. Conclusions

This review summarizes all the known neurological CACNA1C associated manifestations and expands the related phenotype through a case report of a novel de novo mutation that shows additional neurological manifestations (i.e., apraxia). The study also highlights that not always the neurological symptoms without cardiac conduction defects are completely isolated. In fact, other common manifestations of the Timothy syndrome (e.g., dental and gingival defects) may be present in absence of cardiac involvement.

## Figures and Tables

**Figure 1 children-11-00541-f001:**
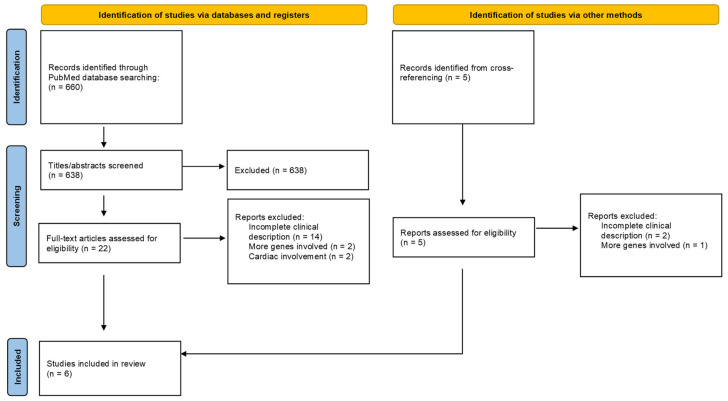
PRISMA flowchart.

**Figure 2 children-11-00541-f002:**
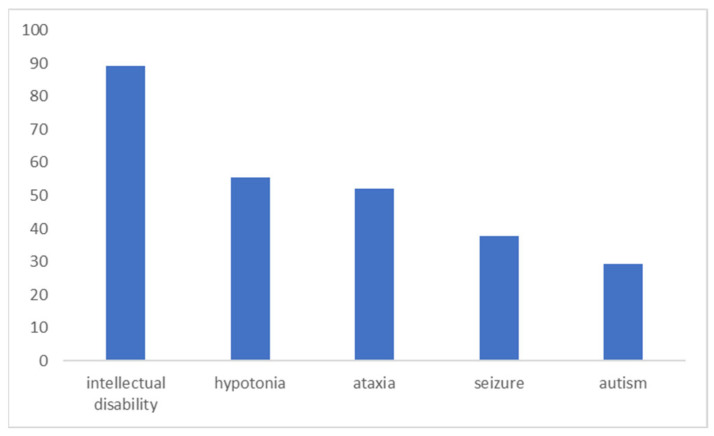
Frequency of neuropsychiatric manifestations in CACNA1C neurologically predominant presentations.

**Figure 3 children-11-00541-f003:**
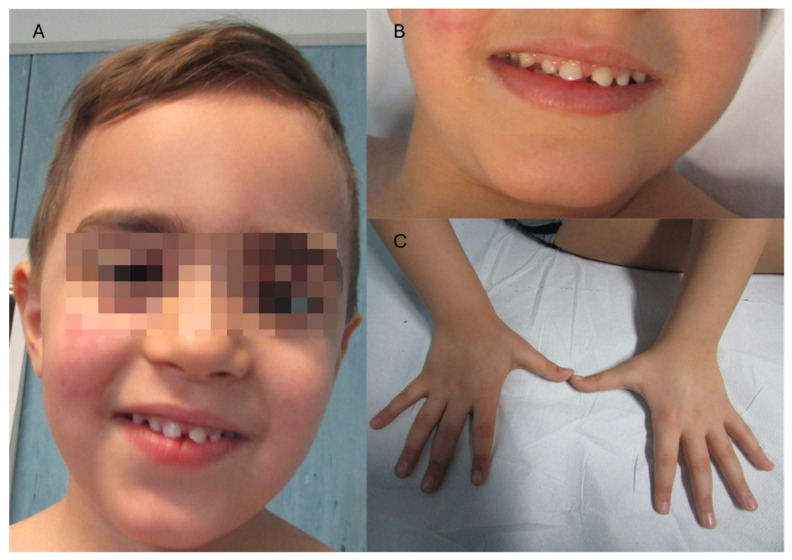
Clinical characteristics of our patient. (**A**,**B**) panels show gingival hyperplasia with abnormal shaped teeth (conical) and diastema. Syndactyly was not present (**C**).

**Table 1 children-11-00541-t001:** Main features of the included studies.

Author, Year	Study Type	Sample Size *	Gender M/F	Mutation Type	Zygosity	Inheritance	ID (tot.)	Seizure (tot.)	Ataxia (tot.)
our patient	CR	1	1/0	missense	Het	dn	1 (1)	0 (1)	0 (1)
[9]	CS	3	2/1	missense	Het	dn	3 (3)	0 (3)	0 (3)
[11]	CS	22	12/10	missense (11)/Truncating (11)	Het	15dn/4pat	18 (22)	9 (22)	8 (22)
[24]	CS	5	5/0	intronic	Het	pat	0 (5)	0 (5)	5 (5)
[26]	CS	3	1/2	missense/intronic	Het	1dn/1pat	1 (3)	2 (3)	0 (3)
[27]	CS	1	1/0	CNV	Het	unk	1 (1)	0 (1)	0 (1)
[28]	CS	1	1/0	CNV	Het	dn	1 (1)	0 (1)	0 (1)

Abbreviations: ID: intellectual disability; CNV: copy number variant; CR: case report; CS: case series; dn: de novo; pat: paternal; unk: unknown; tot.: total. * patients with CACNA1C mutation selected according to inclusion/exclusion criteria.

## Data Availability

The data that support the findings of this study are available from the corresponding author, C.P., upon reasonable request.

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
