# Peer review of "Expanding the Phenotype of the CACNA1C-Associated Neurological Disorders in Children: Systematic Literature Review and Description of a Novel Mutation"

_children, 2024, doi:10.3390/children11050541_

Round 1
Reviewer 1 Report
Comments and Suggestions for Authors
Here are my comments and suggestions:
Introduction
- The structure of the introduction could be more organized. It would be helpful to start with a general background on L-type calcium channels and their functions, then narrow down to the specific gene of interest (CACNA1C) and its associated disorders.
- The authors mention that pathogenic variants in CACNA1C have been related to psychiatric disorders and neurological features, but they don't provide specific examples of these disorders and features. Providing a few examples would help the reader better understand the scope of the problem.
- The authors mention the study by Levy et al. on the neuropsychiatric presentation of CACNA1C-related disorders, but they don't provide any details on the findings of this study. A brief summary of the key results would be helpful.
Methods
- The search terms used for the literature search are not comprehensive enough. The authors only used a few general terms related to neurology and psychiatry, but didn't include specific terms for the various neurological and psychiatric disorders that have been associated with CACNA1C mutations (e.g. schizophrenia, bipolar disorder, ataxia, etc.). Including these specific terms would help ensure that all relevant studies are captured.
- For the genetic analysis, the authors provide a detailed description of the methods used, but don't specify some important details such as the coverage depth and quality thresholds used for variant calling. They also don't mention whether any confirmatory testing (e.g. Sanger sequencing) was performed for the identified variant.
- The authors classify the identified variant as likely pathogenic based on ACMG criteria and CADD score, but don't provide the specific evidence for each ACMG criterion. It would be helpful to know which specific lines of evidence support the pathogenicity of the variant.
Results
- The authors provide a PRISMA flowchart for the study selection process, which is a helpful addition. However, they don't provide the number of studies excluded at each stage of the screening process, which makes it difficult to assess the overall yield of the literature search.
- The authors mention that 35 patients from five studies were included in the systematic review, but they don't provide a table summarizing the key characteristics of these studies (e.g. study design, sample size, patient demographics, mutation types, clinical features). This information would be helpful for the reader to assess the quality and generalizability of the evidence.
- The authors provide a figure showing the frequency of neuropsychiatric manifestations in the included patients, but they don't provide any statistical analysis or comparison to a control group. It's unclear whether these frequencies are significantly different from what would be expected in the general population or in patients with other neurological disorders.
- For the case report, the authors provide a detailed description of the patient's clinical features and genetic testing results. However, they don't provide any information on the patient's developmental history or family history, which would be helpful for understanding the context of the case.
- The authors mention that the identified variant was classified as likely pathogenic based on ACMG criteria and CADD score, but they don't provide the specific evidence for each ACMG criterion. It would be helpful to know which specific lines of evidence support the pathogenicity of the variant.
- The authors mention that the patient underwent extensive genetic testing prior to the clinical exome sequencing, but they don't provide the results of these tests. It would be helpful to know whether any other potentially relevant variants were identified in these previous tests.
Discussion
- The authors highlight the high prevalence of intellectual disability in CACNA1C-related disorders and provide some background on the role of calcium channels in brain development. However, they don't discuss how the specific types of CACNA1C mutations (e.g. missense vs. truncating) or their location within the gene might influence the severity of intellectual disability or other neurological features.
- The authors discuss the potential role of CACNA1C in ataxia, epilepsy, and autism spectrum disorders, but the evidence they provide is somewhat limited. For ataxia, they mention that CACNA1C is expressed in the cerebellum and that calcium channels are reduced in mouse models of ataxia, but they don't provide any direct evidence linking CACNA1C mutations to ataxia in humans. Similarly, for epilepsy and autism, they mention some general associations between calcium channels and these disorders, but don't provide any specific evidence related to CACNA1C.
- The authors highlight the presence of apraxia in their case report and note that this feature has not been commonly reported in previous descriptions of CACNA1C-related disorders. However, they don't provide any further discussion of the potential mechanisms by which CACNA1C mutations could lead to apraxia or how this might relate to the other neurological features.
- The authors mention that their case showed dental and gingival abnormalities, which have previously been reported in Timothy syndrome but not in isolated neurological CACNA1C disorders. This is an interesting observation, but the authors don't provide any further discussion of the potential significance of this finding or how it might relate to the underlying pathophysiology.
- The authors discuss the potential role of alternative splicing and RNA editing in generating multiple CACNA1C isoforms with different functions, and suggest that this could explain the clinical variability of CACNA1C-related disorders. This is an important point, but the authors don't provide any specific examples of how different isoforms might relate to different phenotypes, or discuss any previous literature on this topic.
- The discussion section does not include any mention of the limitations of the current study, such as the small sample size of the systematic review, the potential for publication bias, or the lack of functional validation for the identified variant. Acknowledging these limitations would help provide a more balanced interpretation of the findings.
Reviewer 2 Report
Comments and Suggestions for Authors
the topic is very interesting but you hacee to be more precise, perform a complete systematic review (regist in PROSPERO, offer qualitative datra sinthesys, contrast data, quality assesment table, possible meta-analysis and discuss results) or follow a case report design following CARE checklist and with approval of ethic comintee
Reviewer 3 Report
Comments and Suggestions for Authors
the manuscript titled "Expanding the phenotype of the CACNA1C-associated neurological disorders in children," the paper introduces a novel CACNA1C mutation related to neurological disorders but it lacks sufficient discussion on how this can influence the current diagnostic and therapeutic approaches
the authors have done a thorough review but the integration of findings into a wider context is not good enough, they should do more detailed comparison and analysis of the findings relative to they current literature, concentrating on the neurological presentations without cardiac defects.
the authors should address the potential biases in the study selection and data extraction.
the authors should do a deeper analysis of pathogenicity of the mutation from its impact at the molecular level and its potential interaction with other genetic and environmental factors.
expand the discussion more on the potential clinical implications and future research directions.
Round 2
Reviewer 1 Report
Comments and Suggestions for Authors
The authors responded to my comments very well. Thank you.
Reviewer 2 Report
Comments and Suggestions for Authors
the research is very interesting, if you do a case report, please, follow CARE checklist
Reviewer 3 Report
Comments and Suggestions for Authors
The authors addressed the comments properly.